# DA-KD: Difficulty-Aware Knowledge Distillation for Efficient Large Language Models

Changyi He [1 2]   Yifu Ding [1 2]   Jinyang Guo [1 3]   Ruihao Gong [4]   Haotong Qin [5]   Xianglong Liu [1 2]

## Abstract

Although knowledge distillation (KD) is an effective approach to improve the performance of a smaller LLM (i.e., the student model) by transferring knowledge from a large LLM (i.e., the teacher model), it still suffers from high training cost. Existing LLM distillation methods ignore the difficulty difference among different samples, making the distillation of easy samples unnecessary. This leads to high distillation cost. In this paper, we propose difficulty-aware knowledge distillation (DA-KD) framework for efficient knowledge distillation, in which we dynamically adjust the distillation dataset based on the difficulty of samples. We further observe existing KD loss cannot perform well when most of samples are difficult in the distillation dataset because of unstable optimization and the neglect of hard samples. Therefore, we also propose a new KD loss called bidirectional discrepancy loss (BDL) for effective KD. Extensive experiments demonstrate that our DA-KD framework is effective and efficient. Without bells and whistles, DA-KD can outperform existing state-of-the-art KD methods by 2% with half training cost and even surpass the teacher model with $4.7\times$ compression.

## 1. Introduction

Recent advancements in Large Language Models (LLMs) such as LLaMA and Qwen (Touvron et al., 2023; Bai et al., 2023; Yang et al., 2024a) have garnered significant attention due to their remarkable capabilities and intelligence. However, these models impose substantial computational

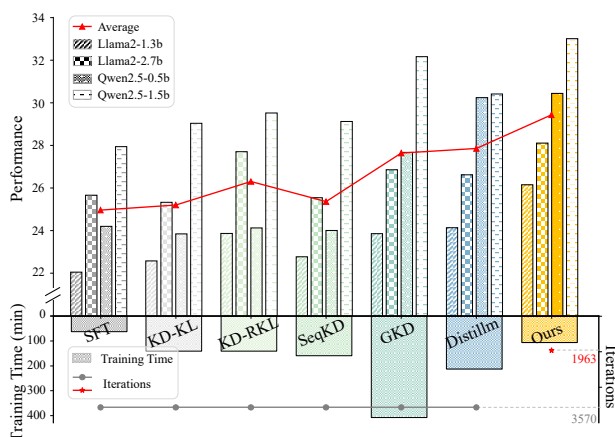

*Figure 1.* Performance of various distillation methods on Dolly dataset in the upper part, and their training time (minutes) and iterations in the bottom part.

and storage requirements, posing challenges for practical deployment (Yang et al., 2024b). To address this issue, numerous model compression methods have emerged, such as quantization (Guo et al., 2024; Lv et al., 2024), pruning (Wang et al., 2024b; Guo et al., 2022; 2023b) and knowledge distillation (Hinton, 2015). Knowledge distillation is an effective approach for constructing smaller and more efficient neural networks. It aims to transfer knowledge from a high-performing teacher model to a compact student model for efficient inference.

Despite the success of KD for effective training of student models, it still suffers from high training cost, which normally requires hundreds of GPU hours when distilling models with billions of parameters (Agarwal et al., 2023). To solve this problem, many efficient distillation approaches were proposed in recent years. They use knowledge caching (Jin et al., 2024; Zheng et al., 2021; Huang et al., 2024), self-distillation (Dong et al., 2023; Zhang et al., 2021; Naeem et al., 2025), dataset condensation (Zhao et al., 2020; Yin et al., 2024) and so on. However, these approaches focus on traditional downstream tasks such as computer vision or language processing, and few of them have delved deeply into efficient distillation using dataset se-

[1] State Key Laboratory of Complex & Critical Software Environment, Beihang University [2] School of Computer Science and Engineering, Beihang University [3] School of Artificial Intelligence, Beihang University [4] SenseTime Research [5] ETH Zurich. Correspondence to: Jinyang Guo <jinyangguo@buaa.edu.cn>.

*Proceedings of the 42nd International Conference on Machine Learning*, Vancouver, Canada. PMLR 267, 2025. Copyright 2025 by the author(s).

lection for generative large models. As a result, it is still an open problem on how to effectively select informative samples for distillation, which makes the efficient distillation for generative large models a non-trivial task.

To solve the aforementioned problems, in this paper, we propose a knowledge distillation framework called **Difficulty-Aware Knowledge Distillation (DA-KD)**, which is designed for efficient distillation of large language models (LLMs). To achieve efficient distillation, we first construct a **Difficult-Aware Data Updating (DiffUp)** strategy, which includes a Distillation Difficulty Score (DDS) to measure the sample complexity based on the performance gap between the teacher and student. As a result, we use DDS to filter out easy samples for condensed data volume. To strengthen data diversity and mitigate the forgetting problem when only difficult samples are adopt (Jiang et al., 2023), we also propose a Stratified Data Updating (SDU) strategy to improve the diversity of the dataset by gradually mixing samples with various DDS.

Furthermore, as we select difficult samples in the distillation process, we find the existing KD loss cannot provide robust optimization for the student. When extreme distribution occurs in the student, existing KD loss will encounter the explosion or vanishing gradient problem. Moreover, we also find it is challenging for existing KD loss to pay more attention to hard samples. To this end, we propose a new loss function called **Bidirectional Discrepancy Loss (BDL)** to restrict the gradient without explosion or vanishing. Specifically, our BDL is built upon traditional KL divergence, but further incorporates combined probability distribution from both teacher and student. From our detailed analysis, we show that our BDL can not only effectively stabilize the optimization for the student but also enforce the distillation to pay more attention to the hard samples.

Our main contributions are summarized as follows:

▶ We propose difficulty-aware knowledge distillation (DA-KD) for efficient and effective LLM distillation,
▶ We also propose difficulty-aware data updating strategy to dynamically update the distillation dataset, which consists of a distillation difficult score for sample difficulty measurement and a stratified sampling strategy for data diversity.
▶ We propose a new loss function called bidirectional discrepancy loss, which can effectively stabilize the student training from difficult samples.
▶ Extensive experiments on multiple benchmark datasets demonstrate the effectiveness of our DA-KD framework.

## 2. Related Work

**Large language model.** Recent advancements in artificial intelligence have garnered significant attention (Tao et al., 2022; 2025b). In particular, Large language models have emerged as transformative advancements in natural language processing, achieving state-of-the-art performance across a broad range of complex tasks. Prior works such as GPT (Radford, 2018) introduced the use of stacked transformer decoders, laying the foundation for subsequent innovations. Building on this, models like the Llama (Touvron et al., 2023; Dubey et al., 2024) and the Qwen (Yang et al., 2024a) series refined transformer architecture to improve efficiency and performance. Other approaches (Guo et al., 2023a; 2021; 2020) were also proposed in recent years by improving either training techniques or network architecture.

Despite these advancements, LLMs often require substantial computational resources due to their massive parameter sizes. In this work, we aim to train an efficient student model by using knowledge distillation technique.

**Knowledge distillation for LLM.** Knowledge distillation transfers knowledge from a large teacher model to a smaller student model (Hinton, 2015) for faster inference. Recently, many knowledge distillation works were proposed for LLM. For example, MiniLLM (Gu et al., 2024) proposed reverse KL divergence (RKL) as the loss function to prevent the student from overestimating the low-probability regions of the teacher. GKD (Agarwal et al., 2023) additionally introduced generalized Jensen-Shannon divergence (JSD), and trained the student on its self-generated outputs to mitigate the training-inference mismatch. DistiLLM (Ko et al., 2024) proposed skew KL divergence (SKL) and skew reverse KL divergence (SRKL) that using a mixture of probability distributions to improve the optimization stability.

However, current KD methods for LLM require a significant amount of training time (Xu et al., 2024; Ko et al., 2024), primarily due to the large-scale training data. In contrast, our DA-KD framework aims at not only effective but also efficient LLM knowledge distillation.

**Data selection for LLM training.** Data selection is increasingly recognized as crucial for optimizing the LLM fine-tuning process (Zhou et al., 2024; Wang et al., 2024a). These methods aim to prioritize subsets of data that are most valuable for training, thus reducing computational costs while preserving or even improving model performance (Tao et al., 2025a). Several recent works have explored data selection for LLM fine-tuning. For example, Chen et al. (2023) introduced Alpagasus to directly leverage ChatGPT for rating each data sample, and select samples with higher ratings. Cao et al. (2023) proposed a linear rule-based approach named Instruction mining by training a linear function to identify the most effective data samples. Li et al. (2023) proposed the IFD method, where a LLM first learns from a small subset of data to acquire foundational capabilities and then uses this learned model to rate and select data from the original dataset.

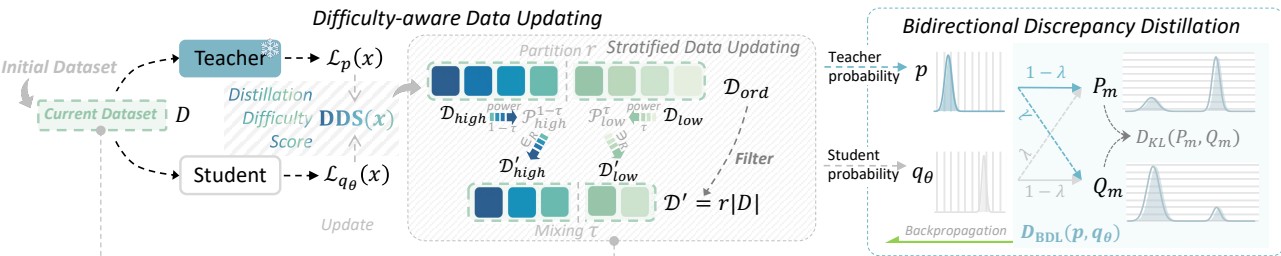

*Figure 2.* Overview of our Difficulty-Aware Knowledge Distillation framework.

**Data selection for knowledge distillation.** Several studies also explored data selection methods in knowledge distillation. For instance, Li et al. (2021) proposed an uncertainty-based data selection method for knowledge distillation of LLMs, while Lan et al. (2025) also introduced a data selection approach specifically designed for knowledge distillation. However these methods are either solely based on student model (Li et al., 2021) or solely based on the teacher model (Lan et al., 2025), ignoring their discrepancy. Different from these works, we propose DA-KD framework to dynamically adjust distillation dataset in the distillation process based on the discrepancy between teacher and student.

## 3. Difficulty-Aware Knowledge Distillation

### 3.1. Overview

Figure 2 shows the overview of our DA-KD framework. Given a large teacher LLM and a small student LLM, we first calculate the cross-entropy loss of each sample in the initial dataset, based on which we calculate the distillation difficult score (DDS) for each sample. Then, we perform the stratified data updating based on the calculated DDS, in which the distillation dataset will be progressively reduced in this process. In the distillation procedure, we use our bidirectional discrepancy loss as the loss function, which can provide stable optimization and pay more attention to hard samples. We will introduce each component in our DA-KD framework in detail.

### 3.2. Difficulty-aware Data Updating

Knowledge distillation for LLMs often involves training a smaller student model to emulate the teacher model's behavior. However, it is inefficient to use the entire dataset indiscriminately for distillation. To address this, we propose **Difficulty-aware data Updating (DiffUp)** to dynamically updates the distillation data based on the sample difficulty to reduce the computational costs of distillation.

#### 3.2.1. DISTILLATION DIFFICULTY SCORE

To evaluate the difficulty of sample $x$ quantitatively, we first propose a new difficulty metric called **Distillation Difficulty Score (DDS)** as the ratio of the loss from student over that from teacher:

$$\text{DDS}(x) = \frac{\mathcal{L}_{q_\theta}(x)}{\mathcal{L}_p(x)}, \quad (1)$$

where $\mathcal{L}_{q_\theta}(x)$ and $\mathcal{L}_p(x)$ represent the cross-entropy loss between the sample $x$ and the ground-truth from the student and teacher models, respectively.

This simple but effective DDS captures the distillation difficulty based on the discrepancy between the teacher and student models. Figure 3 gives examples for different cases. When the student exhibits a high $\mathcal{L}_{q_\theta}(x)$ on a sample but the teacher has low loss $\mathcal{L}_p(x)$ on it, the DDS value becomes large. This situation indicates that the teacher is confident about the sample, whereas the student struggles to do so. Such case highlights the student model requires additional supervision from the teacher. Conversely, the DDS value remains small when both the student and teacher achieve low losses, which means the sample is well-learned by both models, making further distillation unnecessary. When both the teacher and student incur high losses, the DDS value remains small as well, which means the teacher's understanding of the sample is inadequate, limiting its ability to provide meaningful guidance. The essence of DDS is analogous to real-world teaching: effective teaching occurs when concentrating on difficult knowledge that the teacher understands well but the student finds challenging.

#### 3.2.2. STRATIFIED DATA UPDATING

In the distillation process, we propose **Stratified Data Updating** strategy to dynamically select the data in each epoch based on the aforementioned DDS. The core idea is to prioritize difficult samples with high DDS value and gradually reduce the overall data volume in the next epoch for efficient distillation. Suppose we have a training dataset of size $N$, with the data selection ratio per epoch denoted by $r$. This implies that in each epoch, we actually select and use $rN$ data samples for distillation. Initially, the selection ratio $r$ is set to 1, meaning all available data is utilized at the beginning.

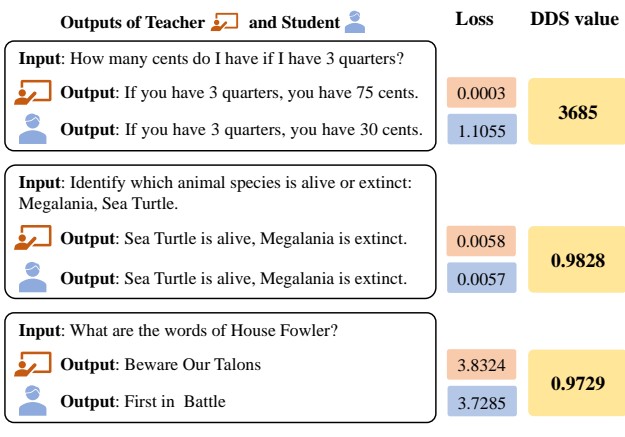

*Figure 3.* Examples of DDS values for different cases.

As distillation progresses, the selection ratio $r$ gradually decreases following a linear or cosine decay schedule. If the total number of epochs is $E$, then the selection ratio $r$ at the $e$-th epoch is:

$$\text{linear decay schedule} : r = 1 - \frac{e}{E},$$
$$\text{cosine decay schedule} : r = \frac{1}{2}\cos\frac{\pi e}{E} + \frac{1}{2} \tag{2}$$

This strategy allows the distillation process to focus on the most valuable data, thereby lowering training costs without sacrificing model performance.

**Stratified sampling for diversity.** Inspired by Jiang et al. (2023), to strengthen data diversity and mitigate the forgetting problem when only difficult samples are adopted, we further propose a stratified sampling strategy to mix various samples when updating distillation data. Specifically, at the beginning of each epoch, we first calculate the DDS for the whole dataset and sort them in descending order. Then, we split the dataset into two partitions by a number threshold $r$, creating low-DDS partition $\mathcal{D}_{low}$ with $1 - r$ percentage of samples and high-DDS partition $\mathcal{D}_{high}$ with $r$ percentage of samples, respectively. Then, we randomly draw samples from both partitions to construct the data subset for distillation for the current epoch.

Formally, let us denote the whole dataset as set $\mathcal{D}$. And the power set $\mathcal{P}(\mathcal{D})$ represents all subsets of $\mathcal{D}$. It is obvious that $\mathcal{D}_{high}, \mathcal{D}_{low} \in \mathcal{P}(\mathcal{D})$ and $\mathcal{D}_{high} \cup \mathcal{D}_{low} = \mathcal{D}$. Next, $P_{high}^{1-\tau}$ represents the collection of subsets in the power set of $\mathcal{D}_{high}$ that contain $(1 - \tau)|\mathcal{D}_{high}|$ elements, i.e.,

$$\forall S \in P_{high}^{1-\tau}, |S| = (1-\tau)|\mathcal{D}_{high}|, \text{and } S \subseteq \mathcal{D}_{high}, \tag{3}$$

where $|\cdot|$ returns the number of element of the set, and $\tau$ is to balance the selection ratio in high and low partitions. Before the start of each epoch, we compute and sort to obtain $\mathcal{D}_{high}$, and then randomly select an element $\mathcal{D}'_{high}$ from $P_{high}^{1-\tau}$, i.e., $\mathcal{D}'_{high} \in_R P_{high}^{1-\tau}$ and $\mathcal{D}'_{high} \subseteq \mathcal{D}_{high}$, where

$\in_R$ denotes a random selection from a finite set. The same procedure is applied to the $\mathcal{D}_{low}$ set, where an element $\mathcal{D}'_{low}$ is randomly selected from $P_{low}^{\tau}$, i.e., $\mathcal{D}'_{low} \in_R P_{low}^{\tau}$, where $P_{low}^{\tau}$ represents the collection of subsets in the power set of $\mathcal{D}_{low}$ that contain $\tau|\mathcal{D}_{high}|$ elements. Thus, the activated data for this training epoch is $\mathcal{D}' = \mathcal{D}'_{low} \cup \mathcal{D}'_{high}$, and is easy to know that $|\mathcal{D}'_{low}| + |\mathcal{D}'_{high}| = \tau|\mathcal{D}_{high}| + (1 - \tau)|\mathcal{D}_{high}| = r|\mathcal{D}|$.

This stratified sampling strategy ensures the distillation dataset not only prioritizes challenging samples but also retains representative easier examples, which prevents potential biases toward high-DDS samples and allows the student to effectively generalize across the entire data distribution.

### 3.3. Bidirectional Discrepancy Distillation

As we use our DiffUp approach to construct our distillation dataset, the retained training samples are often more challenging, exhibiting greater discrepancies between the teacher and student.

To achieve robust and stable distillation, we propose a new distillation loss function called **Bidirectional Discrepancy Loss (BDL)** building upon KL divergence, but further integrate the two measured probability distributions, which can be formally expressed as:

$$D_{\text{BDL}}(p, q_\theta) =$$
$$D_{\text{KL}}\left(((1-\lambda)p + \lambda q_\theta)\|(\lambda p + (1-\lambda)q_\theta)\right), \tag{4}$$

where $p$ and $q_\theta$ denote the probability distributions from teacher and student, respectively. $\lambda$ is the coefficient to balance the contribution of $p$ and $q_\theta$ during the mixing. $D_{\text{KL}}$ denotes the standard KL divergence.

**Remark.** Our BDL can perform well on difficult samples because it can provide robust optimization for the student. Specifically, let us denote $P_m = (1-\lambda)p + \lambda q_\theta$ and $Q_m = \lambda p + (1-\lambda)q_\theta$, we compute the gradient of BDL w.r.t. student parameter $\theta$:

$$\nabla_\theta D_{\text{BDL}}(p, q_\theta) = \nabla_\theta D_{\text{KL}}(P_m, Q_m)$$
$$= \nabla_\theta \sum_x P_m(x) \log \frac{P_m(x)}{Q_m(x)}$$
$$= \sum_x \underbrace{[\lambda \log \frac{P_m(x)}{Q_m(x)} + \lambda - (1-\lambda)\frac{P_m(x)}{Q_m(x)}]}_{C(x)} \nabla_\theta q_\theta(x),$$
$$\tag{5}$$

where $C(x)$ determines the direction and magnitude of each term in $\nabla_\theta q_\theta(x)$. Next, we substitute $P_m$ and $Q_m$ into $P_m(x)/Q_m(x)$ to analyze the influence of the blend distribution on the gradient:

$$\frac{P_m}{Q_m} = \frac{(1-\lambda)p + \lambda q_\theta}{\lambda p + (1-\lambda)q_\theta}. \tag{6}$$

Assuming that the distribution of $q_\theta$ may be extremely small or large for hard samples. If $q_\theta(x) \to 0$, we have $\frac{P_m(x)}{Q_m(x)} \to \frac{(1-\lambda)p(x)}{\lambda p(x)} = \frac{1-\lambda}{\lambda}$. On the contrary, if $q_\theta(x) \to \infty$, we have $\frac{P_m(x)}{Q_m(x)} \to \frac{\lambda q_\theta(x)}{(1-\lambda)q_\theta(x)} = \frac{\lambda}{1-\lambda}$. A similar phenomenon can be observed when $p(x)$ approaches 0 or $\infty$.

This leads to our first conclusion: BDL stabilizes the training by restricting the gradients without explosion or vanishing. Specifically, the range of $C(x)$ is solely determined by $\lambda$, and is independent of $q_\theta$ or $p$. $C(x)$ falls between $(\lambda \log \frac{1-\lambda}{\lambda} + 2 - \frac{1}{\lambda}$ and $\lambda \log \frac{\lambda}{1-\lambda})$ (see Figure 5 in the appendix for illustration). Therefore, since $\lambda$ is a pre-defined value, the range of $C(x)$ is finite and known in advance. This prevents $C(x)$ from becoming excessively large or small due to $q_\theta$ or $p$ approaching extreme or less active, thereby avoiding gradient explosion or vanishing.

Furthermore, by analyzing the range and monotonicity of $C(x)$ in Appendix A, we derive our second conclusion: BDL emphasizes the samples that have distinct output probabilities by teacher and student model (i.e., difficult samples). Specifically, denoting $z = P_m/Q_m$, then we have

$$\frac{\partial C(x)}{\partial z} = \frac{\lambda}{z} - (1-\lambda). \qquad (7)$$

when $z < \frac{\lambda}{1-\lambda}$, $\frac{\partial C(x)}{\partial z}$ exhibit a positive value, indicating $C(x)$ is monotonically increasing with $z = P_m/Q_m$. From our experiments, $\lambda = 0.9$ excels the best performance. Considering the distribution from both teacher and student are usually small, $z < \frac{0.9}{1-0.9}$ takes most of cases in the distillation. Therefore, $C(x)$ can be treated as monotonically increasing with $z$. From Appendix A, we also demonstrate $z = P_m/Q_m$ is also monotonically increasing with $q_\theta/p$ when $\lambda > 0.5$. As a result, $C(x)$ is monotonically increasing with respect to $q_\theta/p$ in our setting. We empirically observe most difficult samples have $q_\theta \gg p$, so we have larger $C(x)$ on these difficult samples. Although this assumption does not always hold, we find it is helpful for the distillation of our DA-KD. This property can enlarge the gradient of these hard samples, which enforce our BDL to pay more attention on them.

Meanwhile, when $\lambda \in (0, 0.5)$, $\frac{P_m}{Q_m}$ is closer to $\frac{p}{q_\theta}$, and $C(x)$ is constantly negative, making BDL behaves similar to standard KL divergence. When $\lambda > 0.5$, $\frac{P_m}{Q_m}$ transits to $\frac{q_\theta}{p}$, and $C(x)$ takes positive values to make $q_\theta$ increase, making BDL behaves similar to reverse KL divergence. However, $\lambda = 0.9$ excels the best performance when $C(x)$ is either positive or negative, which provides a larger range of $C(x)$, and also brings a bigger $\frac{P_m}{Q_m}$ (see Appendix A). This demonstrates that the combination of KL and reverse KL helps the training and convergence of the student. But a higher $\lambda$ may also lead to aggressiveness and gradient explosion. The complete derivation and proof can be found

**Algorithm 1** Difficulty-aware data updating in our DA-KD

**Input:** Initial selection ratio $r = 1$; the whole dataset $\mathcal{D}$ with size $N = |\mathcal{D}|$; balancing coefficient $\tau$; total training epochs $E$; teacher and student model $\theta_T$ and $\theta_S$.

**Output:** Trained student model $\theta_S$.

1: *Iterate the model using $\mathcal{D}$, and obtain initial DDS*
2: **for** $i \leftarrow 2$ to $E$ **do**
3:     *Update the selection ratio $r$ based on Eq. (2)*
4:     $\mathcal{L}_{q_\theta}(\mathcal{D}), \mathcal{L}_p(\mathcal{D}) \leftarrow \theta_T(\mathcal{D}), \theta_S(\mathcal{D})$
5:     $\text{DDS} \leftarrow \frac{\mathcal{L}_{q_\theta}(\mathcal{D})}{\mathcal{L}_p(\mathcal{D})}$
6:     $\mathcal{D}_{ord} \leftarrow$ *Descending ordered $\mathcal{D}$ with updated DDS*
7:     $\mathcal{D}_{high} \leftarrow \mathcal{D}_{ord}[: rN]$
      $\mathcal{D}_{low} \leftarrow \mathcal{D}_{ord}[rN :]$
8:     $\mathcal{P}_{low}^\tau \leftarrow \forall t \in \mathcal{P}(\mathcal{D}_{low}), s.t.|t| = \tau|\mathcal{D}_{high}|$
      $\mathcal{P}_{high}^{1-\tau} \leftarrow \forall t \in \mathcal{P}(\mathcal{D}_{high}), s.t.|t| = (1-\tau)|\mathcal{D}_{high}|$
9:     $\mathcal{D}_{low}^{'}, \mathcal{D}_{high}^{'} \overset{\in_R}{\leftarrow} \mathcal{P}_{low}^\tau, \mathcal{P}_{high}^{1-\tau}$
10:    $\mathcal{D}' \leftarrow \mathcal{D}_{high}^{'} + \mathcal{D}_{low}^{'}$
11:    $y_T, y_S \leftarrow \theta_T(\mathcal{D}'), \theta_S(\mathcal{D}')$
12:    $\mathcal{L}_{\text{BDL}} \leftarrow D_{\text{BDL}}(y_T, y_S)$
13:    $\mathcal{L}_{\text{BDL}}.\text{backward}()$
14: **end for**
15: **return** Student model $\theta_S$

in Appendix A.

In a word, by gradient analysis, we claim that BDL not only stabilizes the optimization process inherently, but also emphasizes the hard samples that induce distinct activations. Compared to the traditional KL divergence, the blending of teacher and student distributions in the numerator and denominator results in smoother gradients, avoiding unstable updates in extreme cases where the teacher and student distributions differ significantly. At the same time, the gradients are regularized based on both distributions, emphasizing the importance of hard samples during training. We provide a detailed parameter analysis experiment to verify the behavior of BDL under different $\lambda$ to show that our BDL ensures more stable and efficient update during training.

After using the proposed BDL and the DiffUp methods, we can effectively focus on more difficult but informative samples, and thus achieve efficient distillation. To provide better explanation of the process for our DA-KD framework, we present a description of the procedure in Algorithm 1.

## 4. Experiments

### 4.1. Experiments Setup

**Tasks and Datasets.** To validate the effectiveness of our DA-KD framework, we conduct two categories of experiments: task-agnostic instruction following and task-specific experiments.

*Table 1.* Results of task-agnostic instruction following on Llama2 and Qwen2.5 models. We report the average ROUGE-L scores across five random seeds. The **bold** and underline indicate the best and second-best results, respectively.

| Model | #Params | Method | DollyEval | SelfInst | Super-Natural | Unnatural | VicunaEval | Avg. |
|---|---|---|---|---|---|---|---|---|
| **Llama2** | 7B | Teacher | 29.61 | 20.98 | 34.11 | 35.56 | 19.19 | 27.89 |
| | 2.7B | SFT | 27.52 | 19.48 | 30.93 | 32.09 | 18.24 | 25.65 |
| | | KD-KL (Hinton, 2015) | 27.71 | 19.19 | 30.24 | 31.75 | 17.73 | 25.32 |
| | | KD-RKL (Gu et al., 2024) | 28.60 | **19.74** | 35.18 | 36.26 | 18.71 | 27.70 |
| | | SeqKD (Kim & Rush, 2016) | 27.78 | 18.64 | 30.95 | 32.14 | 18.13 | 25.53 |
| | | GKD (Agarwal et al., 2023) | 28.72 | 19.48 | 33.12 | 33.74 | 19.17 | 26.85 |
| | | Distillm (Ko et al., 2024) | 27.93 | 18.90 | 32.03 | 35.42 | 18.76 | 26.61 |
| | | DA-KD (ours) | **29.04** | 19.50 | **35.29** | **37.39** | **19.34** | **28.11** |
| | 1.3B | SFT | 25.85 | 14.59 | 24.13 | 28.22 | 17.41 | 22.04 |
| | | KD-KL (Hinton, 2015) | 26.17 | 15.13 | 24.97 | 29.22 | 17.34 | 22.57 |
| | | KD-RKL (Gu et al., 2024) | 25.66 | 15.01 | 27.89 | 32.39 | **18.34** | 23.86 |
| | | SeqKD (Kim & Rush, 2016) | 25.98 | 15.00 | 25.36 | 29.83 | 17.66 | 22.76 |
| | | GKD (Agarwal et al., 2023) | 26.55 | 16.51 | 27.88 | 30.63 | 17.66 | 23.85 |
| | | Distillm (Ko et al., 2024) | **27.16** | 16.52 | 27.27 | 31.98 | 17.70 | 24.13 |
| | | DA-KD (ours) | 26.75 | **18.05** | **32.35** | **36.09** | 17.53 | **26.15** |
| **Qwen2.5** | 7B | Teacher | 31.37 | 25.98 | 43.61 | 40.19 | 23.15 | 32.86 |
| | 1.5B | SFT | 28.05 | 21.12 | 34.21 | 37.06 | 19.22 | 27.93 |
| | | KD-KL (Hinton, 2015) | 28.01 | 21.97 | 37.24 | 37.94 | 19.98 | 29.03 |
| | | KD-RKL (Gu et al., 2024) | 28.40 | 20.60 | 37.93 | 39.46 | 21.18 | 29.51 |
| | | SeqKD (Kim & Rush, 2016) | 28.21 | 20.98 | 35.79 | 36.52 | 21.46 | 28.59 |
| | | GKD (Agarwal et al., 2023) | 28.89 | 23.78 | 44.76 | 39.56 | 23.80 | 32.16 |
| | | Distillm (Ko et al., 2024) | **29.38** | 21.74 | 40.55 | 37.41 | 22.96 | 30.41 |
| | | DA-KD (ours) | 28.91 | **24.08** | **45.73** | **42.31** | **24.04** | **33.01** |
| | 0.5B | SFT | 25.98 | 14.89 | 30.74 | 32.82 | 16.54 | 24.19 |
| | | KD-KL (Hinton, 2015) | 24.59 | 16.07 | 30.00 | 30.95 | 17.57 | 23.84 |
| | | KD-RKL (Gu et al., 2024) | 25.09 | 15.30 | 29.06 | 32.01 | 19.08 | 24.11 |
| | | SeqKD (Kim & Rush, 2016) | 26.28 | 17.40 | 29.57 | 31.64 | 18.90 | 24.76 |
| | | GKD (Agarwal et al., 2023) | 26.24 | 18.74 | 38.45 | 33.31 | 21.56 | 27.66 |
| | | Distillm (Ko et al., 2024) | **27.72** | 19.55 | 42.19 | 37.80 | **23.94** | 30.24 |
| | | DA-KD (ours) | 27.36 | **19.74** | **43.63** | **39.52** | 21.97 | **30.44** |

For instruction following experiments, we choose the `databricks-dolly` (Conover et al., 2023) dataset processed by Gu et al. (2024) for distillation. Then, we evaluate the trained student models on five instruction-following datasets: Dolly evaluation (Conover et al., 2023), Self-Instruct (Wang et al., 2022a), Super-Natural Instructions (Wang et al., 2022b), Unnatural Instruction(Honovich et al., 2022) and Vicuna evaluation(Chiang et al., 2023). The evaluation metric is ROUGE-L (Lin, 2004). To mitigate the randomness, we report the average ROUGE-L score under five different random seeds.

For task-specific experiments, we consider two distinct tasks for evaluation: text summarization using SAM-Sum (Gliwa et al., 2019), and mathematical reasoning with GSM8K (Cobbe et al., 2021). We use ROUGE-L score and zero-shot accuracy for measuring, respectively.

**Models.** We evaluate our DA-KD using multiple teacher-student model pairs. For task-agnostic instruction fol-

lowing, we employ LLaMA2-7B (Touvron et al., 2023) and Qwen2.5-7B (Yang et al., 2024a) as teacher models, with Sheared-LLaMA2-2.7B/1.3B (Xia et al., 2023) and Qwen2.5-1.5B/0.5B as their respective students. For task-specific experiments, we utilize LLaMA2-7B and Qwen2.5-7B as teachers, distilling knowledge into LLaMA2-1.3B and Qwen2.5-1.5B, respectively. Additionally, we include LLaMA3.2-3B (Dubey et al., 2024) as the teacher model with LLaMA3.2-1B as the student.

**Implementation details.** We train all models for 10 epochs using a batch size of 8. We use the AdamW optimizer and a cosine learning rate scheduler. The initial learning rate is set as 1e-5. In our implementation, we set $\tau$ and $\lambda$ as 0.1 and 0.9, respectively. We use a cosine decay schedule to gradually reduce the data selection ratio $r$.

**Comparison Methods.** We compare our approach with six methods: SFT, KD-KL (Hinton, 2015), KD-RKL (Gu et al., 2024), SeqKD (Kim & Rush, 2016), GKD (Agarwal et al.,

2023), Distillm (Ko et al., 2024). SFT means we directly fine-tune the student without distillation.

## 4.2. Main Results

**Task-Agnostic Instruction-Following.** Table 1 presents the results of task-agnostic instruction-following experiments. Our proposed DA-KD method achieves the highest ROUGE-L scores on most of the evaluation datasets, consistently outperforming other methods across different model types and sizes. For instance, in the case of the Llama2-2.7B model, our approach achieves an average ROUGE-L score of 28.11, surpassing the state-of-the-art methods KD-RKL (27.70) and GKD (26.85). Even on the smallest Qwen2.5-0.5B model, our method attains an average ROUGE-L score of 30.44, exceeding Distillm (30.24) and all other baselines.

Notably, both Llama2-2.7B and Qwen2.5-1.5B outperform their respective teacher models in terms of average performance. We hypothesize that this improvement may come from the more informative training datasets brought by our DiffUP approach and the more stable optimization process caused by BDL.

Moreover, we successfully compress the Qwen2.5 model from 7B to 0.5B (14× reduction in disk storage) with only a 2.42 drop in average performance. Furthermore, compressing Qwen2.5 from 7B to 1.5B (4.7× storage reduction) not only preserves performance but even yields a slight improvement. Similar results are observed in the Llama2 experiments. It further demonstrates the effectiveness of our DA-KD framework, highlighting the potential of our proposed distillation framework in producing smaller large models with minor computational resources and time costs.

**Task-Specific Experiments.** Table 2 reports the results of task-specific experiments conducted on the Llama2-1.3B, Qwen2.5-1.5B and Llama3.2-1B for text summarization and mathematical reasoning. Our DA-KD method achieves the best performance across both tasks, demonstrating its effectiveness in diverse task-specific scenarios. Specifically, for the text summarization task on the SAMSum dataset, our approach consistently surpasses previous state-of-the-art methods. For example, on Llama2-1.3B and Llama3.2-1B, our DA-KD achieves 39.20 and 32.92 respectively, outperforming Distillm (38.73 and 32.53). Notably, on Qwen2.5, the student model distilled by our DA-KD achieves a score of 40.05, surpassing even the 7B teacher model. The remarkable result underscores DA-KD's ability to empower a lightweight student model to retain and even enhance high-quality text generation capabilities.

For the mathematical reasoning task on the GSM8K dataset, our method achieves a zero-shot accuracy of 54.66% on Qwen2.5, 14.56% on Llama2, and 22.37% on Llama3.2, demonstrating significant improvements over competitors.

*Table 2.* Results of task-specific experiments on Llama2-1.3B, Qwen2.5-1.5B and Llama3.2-1B. The **bold** and underline markings indicate the best and second-best results, respectively.

| Model | Method | SAMSum | GSM8K |
|---|---|---|---|
| Llama2 | Teacher (7B) | 40.84 | 38.67 |
| | SFT | 37.11 | 11.52 |
| | KD-KL (Hinton, 2015) | 37.80 | 11.37 |
| | KD-RKL (Gu et al., 2024) | 38.45 | 9.40 |
| | GKD (Agarwal et al., 2023) | 38.39 | 14.18 |
| | Distillm (Ko et al., 2024) | 38.73 | 12.59 |
| | DA-KD (ours) | **39.20** | **14.56** |
| Qwen2.5 | Teacher (7B) | 39.70 | 71.72 |
| | SFT | 37.74 | 41.77 |
| | KD-KL (Hinton, 2015) | 37.91 | 44.50 |
| | KD-RKL (Gu et al., 2024) | 38.75 | 44.88 |
| | GKD (Agarwal et al., 2023) | 38.89 | 50.80 |
| | Distillm (Ko et al., 2024) | 39.21 | 46.70 |
| | DA-KD (ours) | **40.05** | **54.66** |
| Llama3.2 | Teacher (3B) | 33.47 | 37.83 |
| | SFT | 31.78 | 12.59 |
| | KD-KL (Hinton, 2015) | 29.57 | 7.28 |
| | KD-RKL (Gu et al., 2024) | 30.51 | 14.03 |
| | GKD (Agarwal et al., 2023) | 30.43 | 17.82 |
| | Distillm (Ko et al., 2024) | 32.53 | 20.17 |
| | DA-KD (ours) | **32.92** | **22.37** |

*Table 3.* Training efficiency comparison.

| Method | Iterations | Traing time (minutes) |
|---|---|---|
| SFT | 3570 | 62.81 |
| KD-KL (Hinton, 2015) | 3570 | 140.75 |
| KD-RKL (Gu et al., 2024) | 3570 | 141.40 |
| SeqKD (Kim & Rush, 2016) | 3570 | 159.73 |
| GKD (Agarwal et al., 2023) | 3570 | 408.24 |
| Distillm (Ko et al., 2024) | 3570 | 213.34 |
| DA-KD (ours) | **1963** | **106.35** |

However, performance degradation is still observed for smaller models on GSM8K. One possible explanation is the inherent complexity of mathematical reasoning, requiring stronger logical inference and multi-step reasoning capabilities. Consequently, distilling such knowledge into a smaller model is inherently more challenging, and remains an open problem for future work to enhance knowledge transfer in mathematical reasoning.

**Training costs.** As our objective is to develop an effective and efficient distillation framework for large language models, we also evaluate the training efficiency of our method compared to existing work. Table 3 presents the number of training iterations and time required for end-to-end distillation. All the test cases compress Llama2-7B into a 2.7B model using four NVIDIA A800 GPUs.

Table 4. Ablation study of our DiffUp method.

| Method | Dolly | SelfInst | Super-N | Unnatural | Vicuna | Avg. |
|---|---|---|---|---|---|---|
| DiffUp | 26.75 | 18.05 | 32.35 | 36.09 | 17.53 | 26.15 |
| w/o DDS | 26.05 | 16.70 | 31.27 | 33.86 | 17.53 | 25.08 |
| w/o SDU | 26.65 | 17.37 | 31.11 | 34.56 | 16.54 | 25.24 |
| w/o DiffUp | 26.73 | 18.51 | 32.07 | 35.64 | 16.58 | 25.91 |

We take SFT as the baseline performance since it directly fine-tunes the student model on the corresponding dataset without knowledge distillation. Compared to all the other typical and commonly-used distillation frameworks, our DA-KD significantly reduces the computational cost of distillation while maintaining superior performance. Specifically, DA-KD requires only 1,963 training iterations, which is 55% fewer compared to all other distillation methods.

In terms of training time, our DA-KD method requires only 26.1% of the GPU hours compared to GKD (106.35 vs. 408.24 minutes) and 49.9% compared to Distillm (106.35 vs. 213.34 minutes), demonstrating remarkable efficiency compared to other methods. That is because as the training process goes on, we use fewer data with hard samples while filtering out the most simple ones, which brings significant time saving for each single epoch. Methods such as GKD and Distillm incur substantial computational costs because they rely on student-generated outputs during training, which increases processing overhead. Although it will introduce an extra computational cost when computing DDS in DA-KD (for example, the total time of 106.35 minutes in Table 3 consists of 28.31 minutes of DDS computation and 78.04 minutes of distillation), we can achieve a reduction of 55% in training iterations by using DDS to filter the distillation data. As a result, we can still achieve an overall reduction in computational cost compared to existing KD methods.

## 4.3. Ablation Study

In this section, we distill Llama2-7B to 1.3B as an example and conduct extensive ablation studies to validate the effectiveness of each component of our DA-KD approach.

### 4.3.1. EFFECT OF DIFFUP

To verify the effectiveness of our proposed difficulty metric DDS, we first remove each component in DiffUp. The result denoted as "**w/o DDS**" in Table 4 means we randomly select data for distillation at each epoch instead of according to their DDS, showing that our DDS metric can bring 1.07 improvement on the average ROUGE-L score, highlighting the importance of DDS in ordering and selecting challenging samples for distillation. For comparison denoted as "**w/o SDU**", we remove the mixing strategy, making all the samples in the distillation dataset being sampled from the high-DDS partition. So SDU brings 0.91 improvement on the average ROUGE-L score. Finally, when we remove the

Table 5. Comparison results for different distillation loss functions combined with DiffUP. We report the average ROUGE-L scores across five random seeds. The **bold** and underline markings indicate the best and second-best results, respectively.

| Loss | Dolly | SelfInst | Super-N | Unnatural | Vicuna | Avg. |
|---|---|---|---|---|---|---|
| KL | 23.69 | 14.40 | 24.97 | 26.16 | 15.98 | 21.04 |
| RKL | 25.35 | 14.72 | 27.31 | 32.15 | **18.18** | 23.54 |
| JSD | 24.80 | 16.86 | 31.18 | 31.70 | 16.79 | 24.26 |
| SKL | 26.02 | 17.31 | 30.76 | 33.02 | 17.72 | 24.96 |
| SRKL | 26.43 | 16.91 | 29.94 | 34.56 | 17.12 | 24.99 |
| BDL | **26.75** | **18.05** | **32.35** | **36.09** | 17.53 | **26.15** |

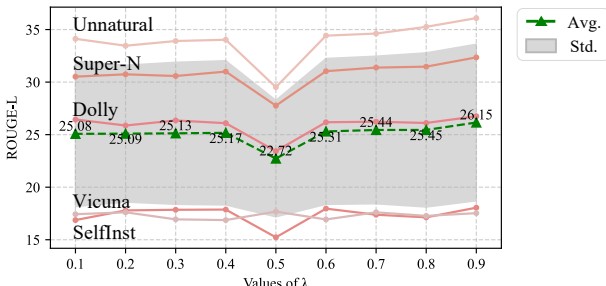

Figure 4. Hyperparameter analysis of $\lambda$ in BDL.

whole DiffUp method and use the entire dataset during distillation process ("**w/o DiffUp**"), the performance degrades 0.24 compared to ours, indicating that dynamically updating the dataset based on the complexity of samples improves the model's performance.

### 4.3.2. EFFECT OF BDL

**Comparisons to Other Distillation Losses.** To verify the effectiveness of our proposed BDL, we conduct experiments by replacing BDL with other objective functions: KL (Hinton, 2015), KL (Gu et al., 2024), JSD (Agarwal et al., 2023), SKL (Ko et al., 2024), and SRKL (Ko et al., 2024). As shown in Table 5, BDL achieves the highest average ROUGE-L score of 26.15, surpassing SRKL by 1.16, and has even larger margin compared to other loss functions, which demonstrates our BDL can stabilize the training and also learn more information in difficult samples.

**Parameter Analysis on $\lambda$.** We conduct the experiment to investigate the performance with different values of $\lambda$ in BDL, and the result is shown in Figure 4, which is consistent to our derivation in Appendix A. When $\lambda = 0.5$, the gradients will have relatively small coefficients which lead to gradient vanishing and performance degradation. When $\lambda \in (0.1, 0.5)$, BDL behaves similar to standard KL divergence and constantly improves the model performance. Moreover, when $\lambda \in (0.6, 0.9)$, it consistently leads to performance gains, and reaches its optimal when $\lambda = 0.9$, which demonstrates that the combination of KL and reverse KL helps the model to converge. The results in Figure 4 confirm the conclusions of our theoretical derivation. However,

determining the optimal $\lambda$, as well as fine-grained tuning of $\lambda$ for the best efficiency, remains an open problem.

## 5. Conclusion

In this paper, we introduce Difficulty-Aware Knowledge Distillation (DA-KD), a novel framework designed to enhance the efficiency and effectiveness of LLM distillation. It includes Difficulty-Aware Data Updating (DiffUp) to filters easy samples using a Distillation Difficulty Score (DDS), and Stratified Data Updating (SDU) to mitigates catastrophic forgetting by mixing samples from all DDS partitions. Additionally, we propose Bidirectional Discrepancy Loss (BDL) to stabilize student training by preventing gradient explosion or vanishing when distilling difficult samples. Extensive experiments demonstrate that DA-KD outperforms state-of-the-art knowledge distillation methods and even surpasses the teacher model in several datasets with only half training cost, making it a highly effective solution for efficient LLM distillation.

One of the limitations of our DA-KD is that its performance depends on the teacher model quality. If the teacher model provides incorrect prediction, the DDS mechanism may misidentify difficult samples, potentially affecting student performance.

## Acknowledgements

This work was supported by the National Science and Technology Major Project (2022ZD0116405), Beijing Municipal Science and Technology Project (No. Z231100010323002), National Natural Science Foundation of China (Nos. 62306025, 92367204, 62476018).

## Impact Statement

This paper presents work whose goal is to advance the field of LLM knowledge distillation. There are many potential societal consequences of our work, none which we feel must be specifically highlighted here.

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

## A. Details about the BDL

### A.1. Proof for BDL

First, we discuss the monotonicity with respect to $q_\theta/p$ for the following equation:

$$\frac{P_m}{Q_m} = \frac{(1-\lambda)p + \lambda q_\theta}{\lambda p + (1-\lambda)q_\theta}, \tag{8}$$

**Proof.** To simplify the analysis, let $x = \frac{q_\theta}{p}$ (representing the relative ratio of $q_\theta$ and $p$). Then we have

$$\frac{P_m}{Q_m} = \frac{(1-\lambda) + \lambda x}{\lambda + (1-\lambda)x}. \tag{9}$$

Next, differentiate $\frac{P_m}{Q_m}$ with respect to $x$:

$$\frac{\partial \frac{P_m}{Q_m}}{\partial x} = \frac{\lambda(\lambda + (1-\lambda)x) - ((1-\lambda) + \lambda x)(1-\lambda)}{(\lambda + (1-\lambda)x)^2} \tag{10}$$

$$= \frac{2\lambda - 1}{(\lambda + (1-\lambda)x)^2}, \tag{11}$$

where the denominator $(\lambda + (1-\lambda)x)^2$ is always positive, and the numerator is a constant $2\lambda - 1$. When $\lambda > 0.5$, $2\lambda - 1 > 0$, which implies $\frac{\partial \frac{P_m}{Q_m}}{\partial x} > 0$, meaning that $\frac{P_m}{Q_m}$ is monotonically increasing with respect to $x$. On the contrary, when $\lambda < 0.5$, $\frac{P_m}{Q_m}$ is monotonically decreasing. And when $\lambda = 0.5$, $\frac{P_m}{Q_m}$ is a constant. Due to the monotonic function of Eq. (9), it is easy to find out the boundary values of $\frac{P_m}{Q_m}$. As $x \to 0$ (i.e., $q_\theta \gg p$), $\frac{P_m}{Q_m} \to \frac{\lambda}{1-\lambda}$; while as $x \to \infty$ (i.e., $q_\theta \ll p$), $\frac{P_m}{Q_m} \to \frac{1-\lambda}{\lambda}$.

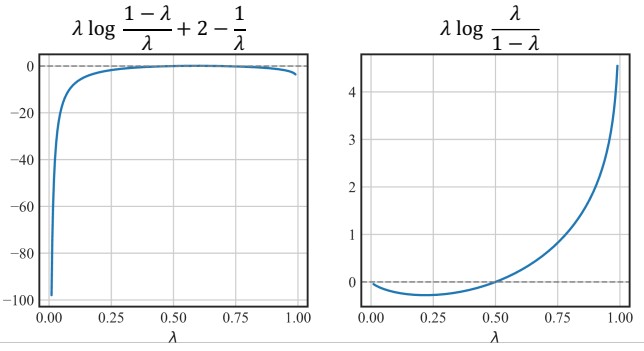

*Figure 5.* Curves of the two boundaries of $C(x)$.

Next, we prove the monotonicity of $C(x)$ in Eq. (5) that

$$C(x) = \lambda \log \frac{P_m(x)}{Q_m(x)} + \lambda - (1-\lambda)\frac{P_m(x)}{Q_m(x)} \tag{12}$$

**Proof.** We first differentiate $C(x)$ as

$$\frac{\partial C(x)}{\partial z} = \frac{\lambda}{z} - (1-\lambda), \tag{13}$$

where $z = P_m/Q_m$. And the second derivative of $C(x)$ is

$$\frac{\partial^2 C(x)}{\partial z^2} = -\frac{\lambda}{z^2}, \tag{14}$$

which constantly below 0 since $\lambda > 0$. Therefore, when $z = \frac{\lambda}{1-\lambda}$, i.e., $\frac{P_m}{Q_m} = \frac{\lambda}{1-\lambda}$, $C(x)$ reaches its maximum $\lambda \log \frac{\lambda}{1-\lambda}$. When $\lambda < 0.5$, $\lambda \log \frac{\lambda}{1-\lambda} < 0$, constantly brings negative $C(x)$. It always makes $Q_m$ to increase and match $P_m$, which is how standard KL Divergence works. However, when $\lambda > 0.5$, $C(x) \in (\lambda \log \frac{1-\lambda}{\lambda} + 2 - \frac{1}{\lambda}, \lambda \log \frac{\lambda}{1-\lambda})$. By drawing the curves of the two boundaries in Figure 5, we can see that when $0.5 < \lambda <\sim 0.7$ (value that around 0.7), $C(x)$ is constantly positive, which is how reversed KL Divergence works in most cases. However, if $\sim 0.7 < \lambda < 1$, the range of $C(x)$ includes both positive and negative values. Since we blend probabilities $q_\theta$ and $p$ into $P_m$ and $Q_m$, it can be regarded as a combination of KL and reversed KL. And it is easy to imagine that when a larger $\lambda$ is applied, the range of $C(x)$ is larger, but if $\lambda$ is close to 1, it behaves similar to reversed KL Divergence.

In a word, we can draw two conclusions. First, given a certain constant $\lambda$, $C(x)$ falls in the range between $\lambda \log \frac{1-\lambda}{\lambda} + 2 - \frac{1}{\lambda}$ and $\lambda \log \frac{\lambda}{1-\lambda}$, which provides finite and known coefficients on the gradients $\nabla_\theta q_\theta(x)$. It avoids the gradient explosion issue during training in case of the outliers in student models' output probability. Second, when the predefined $\lambda \in (0.7, 1)$, $C(x)$ can be positive or negative values, allowing the $q_\theta$ to increase or decrease flexibly. While when being close to 1, it has the probability to give an extreme regularization.

### A.2. Further Discussion on Parameter $\lambda$

Based on above conclusion, we can understand how the loss function BDL functions within the method: when the difference of teacher and student model enlarges regarding to sample $x$, the output probabilities of the two networks differs, leading to a larger $q_\theta/p$. And through the blending of probabilities in BDL, the gradients in the backpropagation will be enhanced by a larger coefficient $C(x)$, which emphasizes the importance of the sample $x$. Based on our analysis above, moderate $\lambda$ is required.

Our parameter analysis experiments in Sec. 4.3.2 show that BDL consistently brings improvements of most $\lambda$ except $0.5$. It is easy to understand that $C(x)$ has the smallest range near 0 when $\lambda = 0.5$, which leads to the gradient vanishing issue. When $\lambda \in (0, 0.5)$, $\frac{P_m}{Q_m}$ is closer to $\frac{p}{q_\theta}$, and $C(x)$ is negative, making BDL behaves similar to standard KL divergence. When $\lambda \in (0.5, \sim 0.7)$, $\frac{P_m}{Q_m}$ transits to $\frac{q_\theta}{p}$, and $C(x)$ takes positive values to make $q_\theta$ increase, making BDL behaves similar to reverse KL divergence. However, $\lambda = 0.9$ excels the best performance, which demonstrates that the combination of KL and reverse KL helps the training and convergence of the model. Meanwhile, a larger $\lambda$ allows a larger range of $C(x)$, and also brings a bigger $\frac{P_m}{Q_m}$. Based on the monotonicity of $\frac{P_m}{Q_m}$ and $C(x)$, we can draw that samples has distinct output probabilities from teacher and student will have a larger $C(x)$ with a larger $\lambda$, which means the gradient will be enlarged and emphasized.

Based on this, we hypothesize that our method inherently addresses some intrinsic issues in existing distillation frameworks, such as stabilizing gradients and emphasizing hard samples. However, determining the optimal value of $\lambda$ remains an open question.

