# OpenReview forum: "DA-KD: Difficulty-Aware Knowledge Distillation for Efficient Large Language Models"
_ICML.cc/2025/Conference — ICML 2025 poster_

### Official Review · Reviewer_nqxA · 2025-03-08

**Overall Recommendation:** 3

**Summary:**

This paper investigates knowledge distillation for large language models (LLMs) and introduces two main contributions: difficulty-aware data updating and bidirectional discrepancy distillation. The difficulty-aware data updating approach estimates sample difficulty using the ratio of cross-entropy losses between the teacher and student models. This information guides a stratified sampling strategy that includes both challenging and easy samples to maintain diverse data representation. The second contribution, bidirectional discrepancy distillation, is a novel loss function that blends teacher and student probabilities, yielding a smoother loss landscape. Together, these components enhance knowledge distillation performance across a variety of scenarios.


## update after rebuttal
Upon reviewing the authors’ rebuttal, I realized that DiffUp was intended to improve efficiency. Part of my initial review was founded on a misunderstanding. I adjusted my score to reflect this.

**Claims And Evidence:**

The paper does not present any theoretical evidence to support its claims, relying instead solely on empirical observations.

While the authors identify difficulty-aware data updating as their primary contribution, the ablation studies indicate that most performance improvements stem from the bidirectional discrepancy distillation. This discrepancy should be addressed in the discussion. Furthermore, since the two components appear to function independently, it may be more beneficial to focus on just one of them.

**Essential References Not Discussed:**

None

**Experimental Designs Or Analyses:**

The experimental setup appears sound. However, including more ablation studies would help clarify the contribution of each component in the algorithm. In its current form, the draft suggests that the loss function plays the primary role in driving performance, while other factors have a comparatively minor impact.

**Methods And Evaluation Criteria:**

The evaluation criteria are reasonable.

**Other Comments Or Suggestions:**

None

**Other Strengths And Weaknesses:**

The BDL loss function does not demonstrate substantial novelty, as there are numerous similar loss functions. Given that the performance improvements largely stem from the loss, the overall novelty of this work is limited.

**Questions For Authors:**

1.	Have you evaluated BDL independently of DiffUp in other scenarios? Have you conducted comparisons with other algorithms?

2.	Have you explored alternative sampling strategies beyond stratified sampling to ensure diversity? I believe there are numerous potential approaches.

3.	Have you examined the inclusion of samples generated by the teacher and the student models?

**Relation To Broader Scientific Literature:**

Smoothing the KL divergence has been explored extensively in prior works across various domains. Including a more in-depth discussion of how BDL compares to these existing KL divergence variants would strengthen this paper.

**Theoretical Claims:**

No theoretical claims are provided.

---

> ### Author Rebuttal · Authors · 2025-04-01
>
> > Q1: The ablation studies suggest most performance improvements stem from BDL.
>
> A1: We need to clarify that DiffUp aims to reduce data and achieve efficiency. The BDL aims to improve performance. It is unfair to only compare accuracy improvement. To verify the effectiveness of DiffUp, we randomly select 55% data and distill with BDL. This setting requires the same number of iterations as DiffUp, and the results show the effectiveness of DiffUP.
> | Method                                    | Dolly | Self  | S-N   | Un    | Vic   | Avg.  |
> | ----------------------------------------- | ----- | ----- | ----- | ----- | ----- | ----- |
> | 55% random data+BDL                     | 24.52 | 15.27 | 29.34 | 32.73 | 16.08 | 23.59 |
> | DiffUP+BDL | 26.75 | 18.05 | 32.35 | 36.09 | 17.53 | 26.15 |
>
> > Q2: Including more ablation studies would help clarify the contribution of each component.
>
> A2: The two components of our DA-KD do not both contribute to performance, which has been discussed in Q1. To verify the contribution of each component, we conduct ablation studies on Llama2-1.3B.
>
> | Method             | Iterations | Dolly | Self  | S-N   | Un    | Vic   | Avg.  |
> | ------------------ | ------------------- | ----- | ----- | ----- | ----- | ----- | ----- |
> | Base       | 3570                | 26.17 | 15.13 | 24.97 | 29.22 | 17.34 | 22.57 |
> | DiffUP             | 1963                | 23.69 | 14.40 | 24.97 | 26.16 | 15.98 | 21.04 |
> | BDL                | 3570                | 26.73 | 18.51 | 32.07 | 35.64 | 16.58 | 25.91 |
> | DiffUP+BDL | 1963                | 26.75 | 18.05 | 32.35 | 36.09 | 17.53 | 26.15 |
>
> We can see that DiffUp reduces the number of training iterations from 3570 to 1963. BDL improves the performance from 22.57 to 25.91. Combining DiffUP and BDL, we can achieve both efficiency and higher performance.
>
> > Q3: Including a discussion of how BDL compares to existing KL variants would strengthen this paper.
>
> A3: Existing KL variants include two main types, mixing KL and RKL or mixing teacher and student distributions.
>
> The former is formalized as a linear combination of KL and RKL [1]. It may suffer from gradient vanishing or explosion when facing extreme distributions. However, our BDL blends teacher and student distributions, avoiding the issues and achieving more stable updating (see Sec. 3.3 of the paper).
>
> The latter category includes SKL [2]. However, it only mixes teacher into student distribution, which may cause gradient vanishing (see Eq. (6) in [2]). However, our BDL also mixes the student into teacher distribution, which can stabilize the gradient for better optimization (see Sec. 3.3 of the paper).
>
> [1] Rethinking kullback-leibler divergence in knowledge distillation for large language models. (2024)
>
> [2] Distillm: Towards streamlined distillation for large language models. (2024)
>
> > Q4: The overall novelty of this work is limited.
>
> A4: Our DA-KD is designed to perform efficient LLM KD. We are happy that other reviewers recognize our contributions as "addresses an important bottleneck in LLM distillation", "simple yet effective" and "well-motivated and appropriate".
>
> > Q5: Have you evaluated BDL independently of DiffUp?
>
> A5: We evaluate BDL independently and compare it with other losses. The results indicate that BDL achieves better performance.
> | Loss | Dolly | Self  | S-N   | Un    | Vic   | Avg.  |
> | ---- | ----- | ----- | ----- | ----- | ----- | ----- |
> | KL   | 26.17 | 15.13 | 24.97 | 29.22 | 17.34 | 22.57 |
> | RKL  | 26.04 | 16.67 | 30.46 | 34.00 | 18.37 | 25.11 |
> | SKL  | 25.87 | 16.89 | 30.42 | 33.66 | 17.47 | 24.86 |
> | SRKL | 26.42 | 17.19 | 31.82 | 35.99 | 16.42 | 25.57 |
> | BDL  | 26.73 | 18.51 | 32.07 | 35.64 | 16.58 | 25.91 |
>
> > Q6: Have you explored alternative sampling strategies?
>
> A6: We have compared with random sampling in Tab 4 of our paper (denoted as w/o DDS). We also compare with the data filtered by the normalized DDS scores as probabilities. Results show stratified sampling performs better.
> | Method                       | Dolly | Self  | S-N   | Un    | Vic   | Avg.  |
> | ---------------------------- | ----- | ----- | ----- | ----- | ----- | ----- |
> | stratified sampling  | 26.75 | 18.05 | 32.35 | 36.09 | 17.53 | 26.15 |
> | sampling based on DDS | 26.62 | 17.48 | 32.21 | 34.96 | 17.51 | 25.76 |
>
> > Q7: Have you examined the inclusion of samples generated by the teacher and the student?
>
> A7: We incorporate teacher-generated or student-generated samples into the distillation. Here are the results. We can see a small performance degradation when using teacher-generated or student-generated samples.
> | Method          |  Dolly | Self  | S-N   | Un    | Vic   | Avg.  |
> | --------------- |  ----- | ----- | ----- | ----- | ----- | ----- |
> | DA-KD           |  26.75 | 18.05 | 32.35 | 36.09 | 17.53 | 26.15 |
> | DA-KD + tea-gen | 26.25 | 17.36 | 31.47 | 35.17 | 16.98 | 25.45 |
> | DA-KD + stu-gen  | 27.27 | 17.91 | 31.01 | 35.01 | 18.11 | 25.86 |

---

### Official Review · Reviewer_ofRn · 2025-03-12

**Overall Recommendation:** 4

**Summary:**

This paper proposes Difficulty-Aware Knowledge Distillation to improve the efficiency of LLM distillation. It proposes a dynamic data updating strategy that leverages a Distillation Difficulty Score to filter out easy samples, and a Bidirectional Discrepancy Loss that stabilizes gradient updates during training. The method is evaluated on instruction-following and task-specific benchmarks, demonstrating superior efficiency and performance compared to state-of-the-art KD methods, with up to 2% improvement in performance and 50% reduction in training cost.



## =================update after rebuttal==============

Thanks for the authors' responses. My concerns have been well addressed.

**Claims And Evidence:**

The authors claim that DA-KD not only improves training efficiency by reducing the data volume and iterations needed, but also enhances performance—even surpassing the teacher model in some scenarios. These claims are supported by comprehensive experiments and ablation studies.

**Essential References Not Discussed:**

The paper has cited most of the relevant literature on LLM KD methods.

**Experimental Designs Or Analyses:**

The experimental design is comprehensive, covering diverse teacher-student pairs and tasks. The ablation studies convincingly demonstrate the contributions of the DiffUp strategy and BDL.

**Methods And Evaluation Criteria:**

The proposed methods are well-motivated and appropriate for addressing the efficiency challenges in LLM distillation. The use of DDS for selective data updating and the stratified sampling strategy are sensible extensions to traditional KD approaches.
The evaluation across multiple benchmarks and metrics (e.g., ROUGE-L, zero-shot accuracy and training time) is solid.

**Other Comments Or Suggestions:**

Discuss potential limitations of DA-KD.

**Other Strengths And Weaknesses:**

Strengths: 1) Novelty: Innovative use of a difficulty-aware data selection mechanism to reduce training cost. 2) Efficiency: Demonstrating reduced computational cost while maintaining/improving performance.


Weakness: See questions.

**Questions For Authors:**

1. The proposed BDL can be converted to KL (when \lambda -> 0) or RKL (when \lambda -> 1). How would it compare to combining KL and RKL linearly, i.e., (1-\lambda)KL+\lambdaRKL ?

2. Is the cross-entropy (CE) loss incorporated when distillation is performed? If so, how is the ratio set?

**Relation To Broader Scientific Literature:**

The work is well situated within the current literature on KD of LLMs. It extends prior approaches (e.g., KD-KL, KD-RKL, GKD, Distillm) by introducing difficulty-aware sample selection and a novel loss function. The connection to data selection in LLM fine-tuning is also clearly articulated.

**Theoretical Claims:**

The paper provides derivations to justify the stability and effectiveness of the bidirectional discrepancy loss. No major correctness issues are found.

---

> ### Author Rebuttal · Authors · 2025-04-01
>
> We sincerely thank you for recognizing our contributions, by stating "well-motivated and appropriate", "experimental design is comprehensive" and "demonstrating reduced computational cost while maintaining/improving performance". Your thoughtful suggestions and questions help a lot to improve our work. We have carefully addressed your questions below:
>
> > Q1: The proposed BDL can be converted to KL (when $\lambda \rightarrow 0$) or RKL (when $\lambda \rightarrow 1$). How would it compare to combining KL and RKL linearly, i.e., $(1-\lambda)KL+\lambda RKL$?
>
> A1: Thank you for your suggestion! We follow your suggestion and perform the experiments. Specifically, we use LLaMA2-7B as the teacher and use LLaMA2-1.3B as the student, and $\lambda$ is set to 0.9. The results are shown below. We observe our BDL performs better than the linear combination of KL and RKL. We hypothesize this is because BDL blends teacher and student distributions, avoiding the issues of gradient explosion or vanishing and achieving a more stable updating process, which is also explained in Sec. 3.3.
>
> | Loss         | Dolly | SelfInst | Super-N | Unnatural | Vicuna | Avg.  |
> | ------------ | ----- | -------- | ------- | --------- | ------ | ----- |
> | BDL          | 26.75 | 18.05    | 32.35   | 36.09     | 17.53  | 26.15 |
> | (1−λ)KL+λRKL | 26.10 | 15.94    | 28.85   | 32.09     | 18.21  | 24.24 |
>
> > Q2： Is the cross-entropy (CE) loss incorporated when distillation is performed? If so, how is the ratio set?
>
> A2：Thank you for your question. The cross-entropy loss is not used in distillation, because we find directly using KD loss performs the best performance. The experiments of loss function combined with cross-entropy loss  are shown below, which is about 3~4% lower than ours. The similar observation was also found in Distillm[1].
>
> [1] "Distillm: Towards streamlined distillation for large language models." arXiv preprint arXiv:2402.03898 (2024).
>
> | Loss              | Dolly | SelfInst | Super-N | Unnatural | Vicuna | Avg.  |
> | ----------------- | ----- | -------- | ------- | --------- | ------ | ----- |
> | 0.0 * CE+1.0 * KD | 26.75 | 18.05    | 32.35   | 36.09     | 17.53  | 26.15 |
> | 0.1 * CE+0.9 * KD | 25.20 | 16.24    | 28.83   | 30.21     | 17.41  | 23.58 |
> | 0.3 * CE+0.7 * KD | 24.37 | 16.27    | 27.01   | 28.28     | 16.44  | 22.48 |
> | 0.5 * CE+0.5 * KD | 24.19 | 15.46    | 26.46   | 28.60     | 16.55  | 22.25 |
>
> > Q3: Discuss potential limitations of DA-KD.
>
> A3: Thanks for your suggestion! One of the limitations of our DA-KD is that its performance is depended on the teacher model quality. If the teacher model provides incorrect prediction, the DDS mechanism may misidentify difficult samples, potentially affecting student performance. We will discuss these limitations in our final version and try to solve it in our future work.

---

> > ### Comment · Reviewer_ofRn · 2025-04-07
> >
> > Thank you for the efforts from authors. I appreciate the authors' detailed rebuttal which has addressed the concerns I raised. I think this work has contribution to the field of LLM KD. I maintain my positive decision.

---

> > > ### Author Response · Authors · 2025-04-07
> > >
> > > Thank you for your recognition of our work and your constructive feedback. We deeply appreciate the time and expertise you have dedicated to evaluating our paper. We will incorporate your suggestions into the final version of the paper.

---

### Official Review · Reviewer_fVGK · 2025-03-12

**Overall Recommendation:** 3

**Summary:**

This paper presents an efficient knowledge distillation method. Specifically, it introduces a Difficulty-aware Data Updating strategy to identify challenging samples and proposes a novel loss function, Bidirectional Discrepancy Distillation, to reduce size of training samples required, thereby improving training efficiency.

**Claims And Evidence:**

Yes, they are.

**Essential References Not Discussed:**

The paper states in the related work section that “the application of data selection methods in knowledge distillation remains unexplored.” However, several studies have already explored this direction. For instance, [4] proposed an uncertainty-based data selection method for knowledge distillation of LLMs, while [5] also introduced a data selection approach specifically designed for knowledge distillation. These existing works should be discussed.

[4]Li, Lei, et al. "Dynamic knowledge distillation for pre-trained language models." 2021.

[5]Lan, Weichao, et al. "Improve Knowledge Distillation via Label Revision and Data Selection.”, 2024.

**Experimental Designs Or Analyses:**

Yes, I did. I checked experiments setup and all the experiments results.

**Methods And Evaluation Criteria:**

Yes, proposed methods do.

**Other Comments Or Suggestions:**

Several typos need to be corrected :

1. “an knowledge distillation ” to “a knowledge distillation” in Line 053.

2. “an LLM first” to “a LLM first” in Line 097.

**Other Strengths And Weaknesses:**

Strengths：

The method is simple yet effective, allowing for a reduction in training time while maintaining performance.The paper is well-written with a clear and logical structure.The authors provide comprehensive experiments to validate the effectiveness of both the data selection strategy and the proposed loss function.

Weakness：

1. In Line 182, the authors state, “we first calculate the DDS for the whole dataset and sort them in descending order.” I would like to clarify whether the computation of DDS and the partitioning of the dataset are performed only once at the beginning of training or updated before each epoch. If it is computed only once, given that DDS values may change as training progresses, how does the method account for this dynamic variation?

2. As mentioned in “Essential References Not Discussed” and “Questions for Authors”, there is a lack of references to and comparisons with some closely related works.

3. The computation of DDS relies solely on the cross-entropy loss between the two models (teacher and student) and the ground truth for a given sample  x . Given this formulation, the data selection method is not inherently limited to LLMs. Would this approach also be effective in other knowledge distillation applications, such as in computer vision models?

**Questions For Authors:**

1. The knowledge distillation method for LLMs proposed in [3] is similar to the approach presented in this paper, as both methods divide samples into two categories based on certain strategies and mix the teacher’s and student’s probability distributions in the loss function. Therefore, Please include an experimental comparison with [3].

2. In Section 3.3, the author stated, “We empirically observe most difficult samples have  q >> p .” Could you explain why the samples selected by DDS exhibit this experimental phenomenon? Is there a theoretical explanation for this observation?

I am willing to raise my score if the authors’ rebuttal addresses my concerns.

[3] Song, Yuncheng, et al. "Self-Evolution Knowledge Distillation for LLM-based Machine Translation." 2024.

**Relation To Broader Scientific Literature:**

This paper proposes a data selection strategy for knowledge distillation (KD) and a new loss function.

The experimental results of this paper are consistent with the conclusions of [1][2], indicating that the model does not require the entire training dataset.

Additionally, the idea of mixing the probability distributions of the teacher and student before computing the KL divergence, as well as the corresponding experimental results, are similar to those in [3].

[1] Zhou, Chunting, et al. "Lima: Less is more for alignment." NeurIPS, 2023.

[2] Li, Ming, et al. "Superfiltering: Weak-to-strong data filtering for fast instruction-tuning.”2024.

[3] Song, Yuncheng, et al. "Self-Evolution Knowledge Distillation for LLM-based Machine Translation." 2024.

**Theoretical Claims:**

Yes, I checked Section 3.3, the remark of the proposed loss function.

---

> ### Author Rebuttal · Authors · 2025-04-01
>
> We sincerely thank you for recognizing the strengths of our work by stating “simple yet effective”, "well-written", and "comprehensive experiments". Your thoughtful suggestions help a lot to improve our work. We have carefully addressed your concerns below:
>
> > Q1: DDS Computation & Dataset Partitioning (Weakness 1)
>
> A1: The DDS computation and dataset partitioning occur before each epoch, dynamically updating difficulty scores based on the student model’s state. While it introduces extra computational cost, we can achieve a 55% reduction in training iterations. As a result, we can still achieve an overall reduction in computational cost compared to existing KD methods, which is shown in Table 3 in our paper.
>
> > Q2: Prior Work on Data Selection in KD (Weakness 2)
>
> A2: Thanks for the suggestion and we apologize for the oversight in our discussion of the related works. Prior data selection methods are either solely based on student model [4] or solely based on the teacher model [5], ignoring their discrepancy. In contrast, our DDS metric integrates both teacher and student distributions. It selects samples that teacher can provide informative guidance to student. We will supplement a detailed discussion in our related work.
>
> > Q3：Generalisability of DDS to other domains (Weakness 3)
>
> A3: To evaluate DDS in a different domain, we conduct experiments on CIFAR-100 using ResNet-50 as the teacher and ResNet-18 as the student. We compare three settings: (1) using the full dataset, (2) randomly selecting 50% of the data, and (3) selecting the top 50% of the data based on DDS. The results  below demonstrate the effectiveness of DDS.
>
> | data                             | accuracy |
> | -------------------------------- | -------- |
> | full training dataset            | 74.80    |
> | random 50% of the data           | 73.66    |
> | top 50% of the data based on DDS | 74.04    |
>
> > Q4: Comparison with SE-KD [3] (Questions 1)
> >
> > [3] Song, Yuncheng, et al. "Self-Evolution Knowledge Distillation for LLM-based Machine Translation." 2024.
>
> A4: SE-KD is conceptually related to our work by both proposing a difficulty metric of the samples and mixing the distributions of teacher and student. However, our method is intrinsically different from this work:
>
> - Different objectives
>
>   SE-KD aims to improve the LLM distillation performance, by dividing samples into two categories and learning with different metrics.
>
>   However, DA-KD designs an efficient KD approach, which gradually reduces the data volume, and thus reduces the computational overhead.
>
> - Different criterion for dividing samples
>
>   SE-KD utilizes KL divergence between the student distribution and a target distribution to quantify the learning difficulty of samples. It does not consider whether a sample is well-learned by teacher.
>
>   However, DA-KD uses the ratio between the student and teacher losses to measure the difficulty of samples. It only selects the samples that are well-learned by the teacher but poorly-learned by the student.
>
> - Different distributions mixture in loss
>
>   SE-KD mixes the student distribution and target distribution unidirectionally, which is similar to the SKL in Distillm[a].
>
>   Differently, DA-KD achieves a bidirectional mixing of teacher and student distributions, leading to a more stable optimization process, which is explained in Sec. 3.3.
>
> We perform an experimental comparison with SE-KD, using LLaMA2-7B (teacher) and LLaMA2-1.3B (student). The results are listed below. Our DA-KD outperforms SE-KD with less training time (about 16% reduction) while obtains better performance.
>
> [a] "Distillm: Towards streamlined distillation for large language models." arXiv:2402.03898 (2024).
>
> | Method | Training Time (mins) | Dolly | Self  | S-N   | Un    | Vic   | Avg.  |
> | ------ | -------------------- | ----- | ----- | ----- | ----- | ----- | ----- |
> | SE-KD  | 86.84                | 25.89 | 17.19 | 32.21 | 36.82 | 16.46 | 25.71 |
> | DA-KD  | 72.12                | 26.75 | 18.05 | 32.35 | 36.09 | 17.53 | 26.15 |
>
> > Q5: Theoretical Explanation of q >> p (Questions 2)
>
> A5: In KD, the teacher model is well-learned and often produces a probability distribution that is smoother and more refined. Conversely, the student, being smaller or less trained, tends to be more overconfident in its incorrect predictions [b], or has less diversity in its output contents [c], resulting in a sharper distribution. This is characterized by high confidence concentrated distribution on fewer options. Therefore, for difficult samples, the probability of the wrong class predicted by student (represented by q) is significantly higher than that predicted by teacher (represented by p).
>
> [b] "On student-teacher deviations in distillation: does it pay to disobey?." NeuIPS 2023.
>
> [c] "Teaching large language models to reason with reinforcement learning." arXiv:2403.04642 (2024).
>
> > Q6: Several typos
>
> A6: Thank you for pointing it out. We will correct all mentioned typos.

---

### Official Review · Reviewer_BqdN · 2025-03-13

**Overall Recommendation:** 4

**Summary:**

The paper introduces DA-KD for knowledge distillation tailored to LLMs. It proposes two main innovations. First, a Difficulty-Aware Data Updating strategy computes Distillation Difficulty Score for each sample, to filter out easy examples and focus training on challenging ones, and employs a Stratified Data Updating strategy to ensure diversity. Second, the paper proposes a new loss function, called Bidirectional Discrepancy Loss, which blends the teacher’s and student’s probability distributions to stabilize gradients and emphasize hard samples. Extensive experiments on both task-agnostic instruction following and task-specific scenarios show that DA-KD outperforms existing state-of-the-art KD methods while reduce the training cost.

**Claims And Evidence:**

The evidence is provided via quantitative comparisons against several baselines on multiple datasets. Overall, the experimental evidence supports the claims.

**Essential References Not Discussed:**

it would benefit from a discussion of alternative methods that focus specifically on dynamic dataset selection in KD. (Revisit the power of vanilla knowledge distillation: from small scale to large scale. Advances in Neural Information Processing Systems 36 (2023): 10170-10183.)

**Experimental Designs Or Analyses:**

Yes.

**Methods And Evaluation Criteria:**

The proposed methods are well motivated. The evaluation criteria (ROUGE-L, accuracy, training time) are appropriate in the context of distillation for LLMs.

**Other Comments Or Suggestions:**

1. It would be helpful to include a discussion on potential limitations.
2. There are some typos or grammar errors:

1) in line 130, “the distillation dataset will progressively reduced” should be “the distillation dataset will be progressively reduced”.
2) in line 161, “only difficult samples are adopt” should be “only difficult samples are adopted”.
3) in the title of Table 5, “disitllation” should be “distillation”.

**Other Strengths And Weaknesses:**

Strength:

The approach addresses an important bottleneck in LLM distillation by reducing training cost.

Weakness:

During the Difficulty-aware Data Updating process, in order to obtain the difficulty scores for each sample, it is needed to calculate the cross-entropy loss of the teacher and student models separately for each sample, which may bring some computational budget.

**Questions For Authors:**

See the weaknesses and comments above.

**Relation To Broader Scientific Literature:**

DA-KD builds on established ideas in KD while incorporating data selection. The framework is positioned as a solution to reduce training cost while maintaining or even improving performance.

**Theoretical Claims:**

The derivations are technical and appear correct. Overall, the theoretical analysis strengthens the motivation for BDL.

---

> ### Author Rebuttal · Authors · 2025-04-01
>
> We sincerely thank you for recognizing the strengths of our work, including “addresses an important bottleneck in LLM distillation” and “experimental evidence supports the claims". Your suggestions are very helpful for improving our paper. We have carefully addressed your concerns below:
>
> > Q1: It would benefit from a discussion of alternative methods that focus specifically on dynamic dataset selection in KD. (Revisit the power of vanilla knowledge distillation: from small scale to large scale. Advances in Neural Information Processing Systems 36 (2023): 10170-10183.)
>
> A1: Thank you for the suggestion. The suggested reference [1] aims to prove that VanillaKD can achieve comparable performance with other KD methods like DKD[2] and DIST[3] when using data augmentation and stronger training strategies instead of data selection. In contrast, we introduce a dynamic dataset selection mechanism to make the distillation process more efficient. We will add the discussion with the work [1] in our final version.
>
> [1] "Revisit the power of vanilla knowledge distillation: from small scale to large scale." Advances in Neural Information Processing Systems 36 (2023): 10170-10183.
>
> [2] "Decoupled knowledge distillation." Proceedings of the IEEE/CVF Conference on computer vision and pattern recognition. 2022.
>
> [3] "Knowledge distillation from a stronger teacher." Advances in Neural Information Processing Systems 35 (2022): 33716-33727.
>
> > Q2: During the Difficulty-aware Data Updating process, in order to obtain the difficulty scores for each sample, it is needed to calculate the cross-entropy loss of the teacher and student models separately for each sample, which may bring some computational budget.
>
> A2: Although it will introduce extra computational cost when computing DDS, we can achieve a 55% reduction in training iterations by using DDS to filter the distillation data (Table 3 in the main paper). As a result, we can still achieve overall reduction in computational cost compared to existing KD methods.
>
> To better analyze the efficiency of our DA-KD framework, we list the training time of different distillation methods when distilling Llama2-7B into a 2.7B. Our DA-KD requires only 26.1% of the training time compared to GKD (106.35 vs. 408.24 minutes) and 49.9% compared to Distillm (106.35 vs. 213.34 minutes), demonstrating the efficiency of the proposed method. We will discuss the data updating cost in our final version.
>
> | Method   | Training time (minutes)                                   |
> | -------- | --------------------------------------------------------- |
> | KD-KL    | 140.75                                                    |
> | KD-RKL   | 141.40                                                    |
> | SeqKD    | 159.73                                                    |
> | GKD      | 408.24                                                    |
> | Distillm | 213.34                                                    |
> | DA-KD    | 106.35 (28.31 for data updating + 78.04 for distillation) |
>
> > Q3: It would be helpful to include a discussion on potential limitations.
>
> A3: Thanks for the suggestion! One of the limitations of our DA-KD is that its performance depends on the teacher model quality. If the teacher model provides incorrect answers, the DDS mechanism may misidentify difficult samples, potentially affecting student performance. We will discuss these limitations in our final version and try to solve it in our future work.
>
> > Q4: There are some typos or grammar errors: ... ...
>
> A4: Thank you for pointing it out. We will correct all mentioned typos and proofread carefully.

---

> > ### Comment · Reviewer_BqdN · 2025-04-07
> >
> > Thank you for your thoughtful rebuttal and clarifications. The authors' additional explanations and insights have further strengthened the paper's contributions.  I maintain my positive accept decision.

---

> > > ### Author Response · Authors · 2025-04-08
> > >
> > > Thank you for your recognition of our work and your constructive feedback. We deeply appreciate the time and expertise you have dedicated to evaluating our paper. We will incorporate your suggestions into the final version of the paper.

---

### Decision · Program_Chairs · 2025-05-01

**Decision:**

Accept (poster)

**Comment:**

This paper presents a well-motivated and empirically validated approach (DA-KD) to tackle the significant challenge of efficient knowledge distillation for LLMs. The core ideas of difficulty-aware data selection (DiffUp/DDS) for efficiency and a tailored loss function (BDL) for effectiveness on hard samples appear sound and demonstrate strong results. The authors provided convincing empirical evidence and effectively addressed reviewer concerns during the rebuttal, including providing additional experiments and clarifications. While one reviewer noted lingering concerns about the BDL's absolute novelty, the overall package and its demonstrated benefits in terms of both efficiency and performance are compelling. The reviewer consensus, with three Accepts and one Weak Accept post-rebuttal, strongly supports acceptance.